# Capecitabine—A “Permanent Mission” in Head and Neck Cancers “War Council”?

**DOI:** 10.3390/jcm11195582

**Published:** 2022-09-23

**Authors:** Camil Ciprian Mireștean, Roxana Irina Iancu, Dragoș Petru Teodor Iancu

**Affiliations:** 1Department of Medical Oncology and Radiotherapy, University of Medicine and Pharmacy Craiova, 200349 Craiova, Romania; 2Department of Surgery, Railways Clinical Hospital, 700506 Iasi, Romania; 3Oral Pathology Department, “Grigore T. Popa” University of Medicine and Pharmacy, 700115 Iasi, Romania; 4Department of Clinical Laboratory, St. Spiridon Emergency Hospital, 700111 Iasi, Romania; 5Department of Medical Oncology and Radiotherapy, “Grigore T. Popa” University of Medicine and Pharmacy, 700115 Iasi, Romania; 6Department of Radiation Oncology, Regional Institute of Oncology, 700483 Iasi, Romania

**Keywords:** HNSCC, HNC, head and neck cancers, capecitabine, 5-FU, chemotherapy, DPD, TP, biomarkers, radiomics, miRNAs

## Abstract

Capecitabine, an oral pro-drug that is metabolized to 5-FU, has been used in clinical practice for more than 20 years, being part of the therapeutic standard for digestive and breast cancers. The use of capecitabine has been evaluated in many trials including cases diagnosed in recurrent or metastatic settings. Induction regimens or a combination with radiation therapy were evaluated in head and neck cancers, but 5-FU still remained the fluoropyrimidine used as a part of the current therapeutic standard. Quantifications of levels or ratios for enzymes are involved in the capecitabine metabolism to 5-FU but are also involved in its conversion and elimination that may lead to discontinuation, dose reduction or escalation of treatment in order to obtain the best therapeutic ratio. These strategies based on biomarkers may be relevant in the context of the implementation of precision oncology. In particular for head and neck cancers, the identification of biomarkers to select possible cases of severe toxicity requiring discontinuation of treatment, including “multi-omics” approaches, evaluate not only serological biomarkers, but also miRNAs, imaging and radiomics which will ensure capecitabine a role in both induction and concomitant or even adjuvant and palliative settings. An approach including routine testing of dihydropyrimidine dehydrogenase (DPD) or even the thymidine phosphorylase (TP)/DPD ratio and the inclusion of miRNAs, imaging and radiomics parameters in multi-omics models will help implement “precision chemotherapy” in HNC, a concept supported by the importance of avoiding interruptions or treatment delays in this type of cancer. The chemosensitivity and prognostic features of HPV-OPC cancers open new horizons for the use of capecitabine in heavily pretreated metastatic cases. Vorinostat and lapatinib are agents that can be associated with capecitabine in future clinical trials to increase the therapeutic ratio.

## 1. Introduction

Head and neck cancers (HNCs) are the sixth most common malignancies in the world, with more than 500,000 new cases occurring each year. Some 90% of all these cancers are squamous cell carcinomas. Even though a multimodal approach with combining treatment methods (surgery, chemotherapy and radiotherapy) is considered optimal, the recurrence rate of 30–50% justifies efforts to identify new therapeutic strategies to improve the HNC prognosis. Chemotherapy brings therapeutic benefits, its role being essential, especially in the locally advanced, recurrent or metastatic disease stages. The administration of less toxic but highly effective chemotherapy regimens is a current concern, although modern therapies such as immune checkpoint inhibitors and monoclonal antibodies have entered the therapeutic arsenal of HNC [1,2].

We conducted a narrative review including significant studies, original articles and abstracts published in the MEDLINE database between 1992 and 2022.

## 2. Capecitabine–More Than 20 Years of Clinical Experience in Different Cancer Types

Used for several decades in therapeutic protocols for gastrointestinal cancers, breast cancer, urinary tract cancers and head and neck cancers, 5-fluoruracil has several disadvantages (reduced bioactivity time in bolus administration) and the necessity to be associated with other chemotherapy agents. This treatment also often requires frequent visits to the hospital, a continuous infusion being given for 5 days, most often through an infusion pump connected through a central venous catheter (CVC). Moreover, the protocol leads to infusion-treatment-related toxicities such as myelo-suppression, gastrointestinal toxicity, the need for dose adjustment, deterioration in quality of life and increased risk of hospital admission. In this context, an oral analogue of 5-fluorouracil is becoming a topic of interest [3,4,5].

Capecitabine is a fluoropyrimidine–carbamate, being included in the therapeutic protocol of metastatic breast cancer and colorectal cancer in combination with other agents or in concurrent treatment with radiotherapy. The principle behind the replacement of fluorouracil with capecitabine focuses on the transformation by enzymes in tumors of this oral pro-drug in 5-FU. Thymidine phosphorylase (TP) is identified in higher amounts at the tumor level; thus the conversion of capecitabine to 5-fluorouracil occurs mainly at the tumor level, resulting in a low concentration of the agent in plasma or normal tissues. In normal and tumor cells, fluorouracil is metabolized by 5-fluoro-2′-deoxyuridine 5′-monophosphate (FdUMP) and 5-fluorouridine tri-phosphate (FUTP). FdUMP is involved in blocking the synthesis of a thymidine triphosphate promoter, with the final consequence of blocking the DNA site. Thus, the reduction of FdUMP levels has the inhibition of cell division as a direct consequence. FUTP can be replaced by uridine tri-phosphate (UTP) during RNA synthesis, resulting in fraudulent RNA [6,7].

## 3. Capecitabine in Head and Neck Cancers—A 5-FU Equivalent Substitute?

With a net benefit of 8% compared to concurrent chemotherapy, chemo–radiation has become a therapeutic standard in locally advanced HNC cases. However, for bulky disease, induction chemotherapy followed by concurrent chemo–radiation is an option often chosen by clinicians, although data on induction chemotherapy are still controversial. The Department of Veterans Affairs Laryngeal Cancer Study Group has since 1991 offered the option of induction chemotherapy followed by radiation therapy as a variant comparable to surgery in its clinical result but with the possibility of preserving the organ in advanced laryngeal cancer. A triple combination in the TPF regimen (docetaxel, cisplatin, 5-fluorouracil) is considered the standard induction regimen, demonstrating benefits over monotherapy or over platinum doublet. However, the TPF regimen is associated with high rates of toxicity, with 31% of patients having quality of life (QOL) affected but an 86% overall response rate and a 3-year survival rate of 65.1%, justifying the use of this regimen in clinical practice. The mTPF is a modified regimen with a favorable toxicity profile, being usable in patients aged >70 years but not eligible for the standard regimen. Therapeutic regimens including platinum salts, 5-fluorouracil and taxanes are among the therapeutic options in recurrent or metastatic disease. EXTREME regimen (fluorouracil/platinum/cetuximab) combines a monoclonal antibodies with chemotherapy. Until first-line immunotherapy was validated as a standard regimen, it was considered the optimal treatment for this category of patients [8,9,10,11,12].

The Phase III study by Custem and collaborators evaluated both the efficacy and toxicity profile of capecitabine in metastatic colorectal cancer, comparing the results with those obtained in a group of patients treated with the standard intravenous fluorouracil/leucovorin protocol (IV 5-FU/LV). Oral capecitabine has been associated with at least equivalent results. The overall toxicity profile was also favorable, with capecitabine being associated with lower grade three-fourths stomatitis and neutropenia, a reduced risk of febrile neutropenia, but an increased incidence of hand–foot syndrome compared to the standard protocol [13].

A Phase II study evaluated the efficacy and tolerability of 1,250 mg/m^2^ of capecitabine twice a day as palliative monotherapy for 1–14 days every 21 days for recurrent or metastatic HNC previously treated with platinum salts. The protocol provides for the administration of at least two cycles, and the overall response rate was 24.2%. The toxicity rate was a maximum of 12.5% (asthenia), 10% for dysphagia, erythrodysesthesia mucositis and 7.5% for diarrhea. The results advocated the inclusion of capecitabine in the palliative treatment of HNC previously treated with platinum salts. Capecitabine monotherapy has shown benefit in recurrent/metastatic nasopharyngeal carcinoma (NPC), according to a study that included 49 patients, 48 of whom were previously treated with platinum-based chemotherapy. With a median follow-up of 10 months, overall survival (OS) at one and two years was 54% and 26%, respectively, and patients who were treated for local–regional recurrences as well as those with hand–foot syndrome had better OS. Péron et al. demonstrated the benefit and feasibility of treatment with capecitabine, and in heavily pretreated frail HNC patients, fatigue, mucositis and hand–foot syndrome were the most commonly reported toxicities [14,15,16].

Won and colleagues in a Phase II study evaluated the efficacy and toxicity of chemotherapy combined with capecitabine and cisplatin (capecitabine 1250 mg/m^2^ twice daily for the first 14 days of a 3-week repeat cycle and cisplatin 60 mg/m^2^ IV day 1) in recurrent or metastatic cases of head and neck squamous cell carcinoma (HNSCC). With an overall survival at 1 year and a survival rate of 10.3 months, respectively, 43.3% of the reported acute grade 3 or 4 toxicities included neutropenia (14.6%), anemia (1.5%), fatigue (4.4%), anorexia (8.8%), diarrhea (4.4%), stomatitis (3.6%) and hand syndrome (1.5%). Moreover, the study did not report toxic deaths related to treatment, and the authors consider the regimen acceptable as a toxicity profile and with a satisfactory therapeutic response [17].

Patients diagnosed with metastatic oropharyngeal cancers associated with human papilloma virus (HPV-OPC) infection have a median overall survival (OS) of over 2 years, being considered eligible to receive multiple palliative therapies. The increased chemosensitivity of this particular subclass of head and neck cancers justifies the proposal by Fazer and colleagues to use capecitabine with possible benefits for heavily pretreated HPV-OPC patients. The average duration of treatment with capecitabine was 9 months in a small group of seven patients, four of them having a partial response, one case showing stationary disease and two patients being diagnosed with progressive disease. It is worth mentioning one case that continued chemotherapy with capecitabine 33 months after the initiation of treatment. The patient selection group was heterogeneous both in terms of initial treatment and metastatic sites. Among the palliative treatments, we mention radiotherapy and ablation of liver metastases, but also biological therapy with cetuximab, immunotherapy with nivolumab and pembrolizumab, as well as multiple chemotherapy protocols including agents such as cisplatin, gemcitabine and pemetrexed. An average time from the diagnosis of metastatic disease to initiation of treatment with capecitabine of 21 months and a median treatment of 9 months with capecitabine-based chemotherapy discontinued for reasons of toxicity or disease progression justifies the authors’ proposal of using capecitabine in heavily pretreated metastatic HPV-OPC cases [18,19]. 

Capecitabine in combination with lapatib has also demonstrated equivalence with the EXTREME regimen in the first line of treatment for metastatic head and neck cancer, other than nasopharyngeal carcinoma, evaluated in patients having an ECOG performance index of 0 to 2, the toxicity profile of the combination being considered favorable. A capecitabine dose of 1000 mg/m^2^) twice daily and lapatinib at a dose of 1250 mg daily with an administration of capecitabine for 14 days of each 21-day protocol included four cycles of chemotherapy associating lapatinib daily until disease progression [20].

Histone deacetylase-inhibitor vorinostat was tested in combination with capecitabine in head and neck cancers, considering the in vivo and in vitro data supporting the activity of vorinostat in combination with deoxy-5-fluorouridine (5′-DFUR) and the potential of vorinostat to upregulate TP. The synergistic antiproliferative result of capecitabine and vorinostat justifies the proposal of Di Gennaro and collaborators to implement clinical trials to support this treatment, the hypothesis formulated more than 10 years ago. Wisniewska-Jarosinska et al. mentioned both an effect of free radicals and an increase in the G0/G1 cell population and reduction of the populations in the S phase as factors that support the cyto- and genotoxic effects in head and neck cancer cells and the protection of healthy cells associated with chemotherapy based on capecitabine [21,22].

Evaluated in a Phase I trial, vorinostat in a maximum tolerated dose (MTD) of 300 mg was administered in cases of Stage III, IVa, IVb HNSCC cancers, including larynx, hypopharynx, nasopharynx, and oropharynx, both HPV positive and negative cases, concurrent with standard chemoradiotherapy. The complete response rate of 96.2% and the favorable toxicity profile, including especially cases of hematological toxicity, justify the testing of vorinostat in Phase II and III trials [23].

## 4. Capecitabine and Radiotherapy

Capecitabine concomitant with accelerated hypo-fractionated radical radiotherapy was evaluated as a possible treatment in locally advanced HNSCC in the study conducted by Jegannathen et al. The proposed dose varied between 450 and 550 mg/m^2^, twice daily for 28 days, with concurrent conformal radiotherapy to a total dose of 55 Gy in 20 fractions delivered in 4 weeks. Treatment was associated with a recurrence rate of 34% and a median OS of 56% at 5 years; 82% of patients completed chemotherapy, and 92% completed radiotherapy. Acute grade 3–4 cardiac, skin and gastrointestinal toxicity related to treatment were also reported; 44% of patients needed a feeding tube, and 6% still needed the device at 1 year. The authors considered treatment as a well-tolerated option with response rates comparable to other concurrent treatment variants [24].

A prospective Phase II study aimed at investigating the safety and efficacy of combined re-irradiation with capecitabine for patients with recurrent HNSCC. The doses of chemotherapy and re-irradiation were 900 mg/m^2^/day and 50 Gy in 25 fractions up to 60 Gy in 30 fractions. With a median time to recurrence after re-irradiation and a response rate of 68% and 19% at a median cumulative dose of 116 Gy, treatment was considered feasible even for patients with a lower Eastern Cooperative Oncology Group (ECOG) performance status, with the most common acute toxicity being mucositis. Thirty-seven patients with stage III or IV HNSCC were included in a study that proposed a concurrent chemoradiotherapy regimen with capecitabine and cisplatin, including a radiotherapy of 1.8–2.0 Gy/fraction, one fraction per day up to a total dose of 70–70.2 Gy. It should be noted that radiotherapy was administered only in the primary tumor, and the study included various anatomical locations of HNC. The partial and complete response rate was 78.4% and 16.2%, respectively; OS and PFS were 76.8% and 57.9%, respectively. The study mentions two cases of grade 3 or 4 neutropenia and one case of febrile neutropenia. In conclusion, the concomitant regimen with cisplatin and capecitabine (two cycles) concurrent with definitive radiotherapy is well tolerated and efficient [25,26].

## 5. Capecitabine in Induction Chemotherapy

The combination of cisplatin and capecitabine and its results were correlated with levels of enzymes and factors identified as potential biomarkers. Thymidine-phosphorylase (TP) expression was increased in the cytoplasm in patients with laryngeal cancer without confirming the results obtained in colorectal cancer studies (TP values correlated with the favorable response in patients treated with capecitabine and irinotecan). Micro-vessel density count, another evaluated parameter, was increased in patients with metastatic disease, the values being lower in cases with localized disease. The results of the study indicate an overall response rate of 68%, of which 39% were complete responses. Moreover, progression-free survival and median overall survival were 6.4 and 12.6 months, respectively [27].

Induction chemotherapy is a preferred option in cases of advanced loco–regional nasopharyngeal carcinoma when added to concomitant chemoradiotherapy. However, the choice of an optimal chemotherapy regimen is still the subject of studies. A multi-centric, randomized Phase 3 trial that included 238 patients from China evaluated an induction regimen with paclitaxel, cisplatin and capecitabine (TPC), comparing results with a cisplatin and fluorouracil (PF) induction regimen. Two cycles of TPC induction (paclitaxel 150 mg/m^2^, cisplatin 60 mg/m^2^ on day 1 and capecitabine 1000 mg/m^2^ twice daily for the first 14 days) were compared with a PF regimen (cisplatin100 mg/m^2^, day 1 and fluorouracil 800 mg/m^2^ daily, days 1–5), both regimens delivered before chemoradiotherapy. The TPC regimen reported failure-free survival at 3 years of about 83.5% compared to 68.9% for the PF regimen. TPC also reduced the risk of distant metastases, and the toxicity profile was comparable to those registered in TPF. With a single death in the PF group, the study demonstrated the benefit and safety profile of the TPC regimen. The maximum tolerated dose (MTD) of capecitabine co-administered with carboplatin and intensity-modulated radiotherapy (IMRT) was assessed for HNSCC stage III/IV. Concomitant radiotherapy with poly-chemotherapy was administered after 6 weeks of induction. MTD was identified as 850 mg/m^2^/day on days 1–14 and 22–35, and myelosuppression was the only significant and dose-limiting toxicity reported. The study designed by Gao et al. investigated the benefit of reducing the toxicity profile for a combined cisplatin–capecitabine induction regimen in NPCs. Data analysis in a group of 136 patients demonstrated the ability of the cisplatin–capecitabine combination regimen delivered in induction settings to reduce HNSCC therapy-associated toxicities [28,29,30,31].

At Roosevelt Hospital in Guatemala during the COVID-19 pandemic, the TPC poly-chemotherapy regimen was tested as a substitute for the standard cisplatin, paclitaxel and 5-FU in order to reduce the time required for hospital treatment. With an average of 3.5 cycles administered, 93.6% of patients responded to treatment, and the rate of discontinuation and dose reduction of therapy was 6.2% and 37.5%, respectively, nausea, vomiting and diarrhea being the main toxicity reported. The authors recommend the initiation of prospective studies of 5-FU substitution with capecitabine [32].

## 6. Capecitabine–From Pharmaco-Dynamics to Biomarkers of Response

An enzyme cascade consisting of three enzymes converts the drug capecitabine into 5-FU with the effect of inhibiting thymidylate synthesis and finally DNA synthesis. Two enzymes (carboxyl-esterase and cytidine-deaminase) are located in the liver tissue, and the final stage of drug metabolism is totally dependent on the presence of TP. TP is identified in a higher amount in the tumor and is a factor associated with angiogenesis and tumor growth and is possibly linked to an unfavorable prognosis. Dihydropyrimidine dehydrogenase (DPD) is the enzyme responsible for converting 5-FU to α-fluoro-β-alanine (FBAL), metabolites that can be eliminated, thus limiting the toxicity associated with chemotherapy. Some 60–90% of capecitabine is converted by this enzyme, while about 10% of the amount of chemotherapeutic is excreted unchanged in the urine. It is estimated that up to 96% of capecitabine is found in the form of FBAL in the urine. Overexpressed or reduced DPD may affect the plasma concentration of the drug and the toxicities associated with capecitabine. The absence of DPD can be potentially fatal due to toxicity associated with the accumulation of large amounts of drugs that cannot be metabolized, but it is very rare [33,34,35].

The prognostic value of this enzyme has been identified in association with first-line therapy with capecitabine in breast cancer and with βIII-tubulin levels and is associated with outcome in cases involving capecitabine and taxanes in first-line chemotherapy. Thymidylate synthase (TS), DPD and TP were evaluated as potential biomarkers in stage III colon cancer patients who received adjuvant chemotherapy in the XELOX protocol or with 5-FU/Leucovorin bolus. A lower tumor level of DPD and a higher TP/DPD ratio have been associated with XELOX regimen efficacy in terms of DFS. In the case of 5-FU/Leucovorin, none of the enzymes appear to have biomarker value. By genotyping the 5′ and 3’ ends, the expression of TS from patients’ peripheral blood before the chemotherapy treatment and by analyzing TS, TP and DPD genes in RNA extracted from a paraffin-embedded tumor, a translational study aimed to evaluate the response to a capecitabine chemotherapy regimen and also to identify the treatment associated toxicities in squamous cell carcinoma of the cervix cases. The analysis only identified the correlation of relative TP tumor expression with anemia; 38 single genes but also metagenes, including cytotoxic cell metagenes have been associated with the benefit of capecitabine adjuvant treatment in TNBC. Both the genes involved in antitumor immunity and those related to the metabolism of capecitabine to 5-FU appear to be involved in the response of this aggressive type of breast cancer to adjuvant chemotherapy with capecitabine [36,37,38,39].

A prospective study included patients diagnosed with colorectal cancer and liver metastases who were treated with oxaliplatin plus 5-fluorouracil or capecitabine in the first line. The study aimed to identify biomarkers associated with PFS and OS, both serological and imagistic biomarkers extracted from magnetic resonance imaging (MRI) being evaluated. The study included 20 cases, and the results demonstrated the ability of chemotherapy to significantly reduce the number of circulating tumor cells. The circulating concentrations of angio-genetic factors were also reduced until they were comparable to pretreatment values. These results were recorded during the second cycle of chemotherapy. No variation of imaging biomarkers were associated with PFS or OS. The infusion/permeability parameter MRI, K-trans, was increased during treatment. The correlation of higher and increasing values of K-trans with OS may be the basis for the use of this imaging parameter as a biomarker for the choice of anti-agiogenic therapy in the protocol that includes chemotherapy based on 5-FU or capecitabine. The NALA Phase III trial looked at the benefit of the combination of neratinib plus capecitabine compared to the combination of lapatinib-capecitabine in previously treated metastatic breast cancer patients. HER2 positive cases were associated with a greater benefit in the neratib–capecitabine combination. PIK3CA mutations were associated in both treatment arms with a shorter PFS. Circulating levels of C-C motif chemokine ligand 5 (CCL5) have been associated with an unfavorable prognosis in pancreatic cancer treated with capecitabine, the result being associated with a possible CCL5-induced immunosuppressive tumor microenvironment [40,41].

Dose adjustment of fluoropyrimidine based on testing for DPD in head and neck cancers was proposed in a prospective study of 65 patients to reduce the rate of toxicity. Monitoring of uracil/UH2 plasma levels was used as a surrogate marker for the enzyme level, and 5-FU doses were reduced depending on the DPD deficiency adapted to clinical parameters. DPD deficiency was mild to marked in 20–28% of cases, with dose reduction ranging from 10% to 100%. In this group, only 9% of patients experienced severe side effects compared to 22% in the group in which no DPD deficiency was tested. There was no need to postpone chemotherapy in the group in which the dose was tested/adjusted. Given the importance of both reducing the rate of severe toxicity and avoiding delayed chemotherapy in head and neck cancers, the authors advocate for routine testing of DPD levels in this group of patients treated with fluoropyrimidines. Adjusting the dose of fluoropyrimidine according to the DPD level, whether it is a reduction or an escalation, is part of the treatment personalization strategy proposed by Chamorey, the correction of the dose of fluoropyrimidine will be directly proportional to the DPD level. High intrinsic lymphocytic DPD activity was assessed as a possible biomarker of PFS, OS and response rate. A cut-off at 0.30 nmol/min/mg protein for DPD activity was identified as significant to discriminate patient outcomes. DPD deficiency is associated with 150 toxic deaths associated with the fluoropyrimidine-based regimen in France annually. In this context, noting a proportion of 50% of 76,200 patients receiving treatment based on this class of agents. Of these, 62%, 26% and 12% representing digestive, breast and neck cancer, respectively, Barin-Le Guellec proposed routine DPD testing for all patients receiving 5-FU or capecitabine [42,43,44].

Radiomics, one of the new branches of artificial intelligence, is based on extracting and analyzing a large volume of data from medical imaging in order to build models with predictive value and prognosis. Radiomics models are currently being evaluated both as independent biomarkers and in multi-omics models to increase diagnostic, predictive, and prognostic accuracy in all areas of oncology. With a growing role in increasing diagnostic accuracy and outcome prediction in oncology, including head and neck cancers, radiomics will be able to play a key role in precision oncology. A study including 118 patients with locally advanced rectal cancer (LARC) demonstrated the ability of a multimodal nomogram model including CT and MRI imaging, radiomics data and clinical data to predict the pathological response to neoadjuvant chemotherapy including fluoropyrimidines. The variation of radiomics (delta radiomics) parameters and the extraction of features from both “tumor-core” and “tumor border” has led to the proposal of “clinic-radiomics” models to predict the response to 5-FU, capecitabine or tegafur-based chemotherapy in LARC [45,46,47,48].

In order to evaluate the discriminating ability of a radiomic signature using features extracted from computed tomography (CT) simulation for radiotherapy planning, a group of researchers from the Universities of Tübingen and Florence proposed two study and validation cohorts involving patients with rectal cancer neoadjuvant treated with chemotherapy based on capecitabine or 5-fluorouracil and concurrent radiotherapy (standard fractionation regimen). The long-course neoadjuvant treatment protocol was the one proposed in the department, and planning the target volumes were defined manually for radiotherapy treatment planning. The radiomic analysis was performed with an in-house-developed software and the degree of tumor regression Dworak was chosen as a surrogate end point. The analyzed data included 1150 radiomic features obtained from a study group of 126 patients in the study cohort and 75 cases in the validation cohort. The study identified the five metafeature complex in association with the five-nearest neighbor model as the most robust and reproducible radiomic model which can predict pre-treatment response to neoadjuvant therapy in rectal cancer. Bordron’s study had the same goal of predicting the response to neoadjuvant treatment based on fluoropyrimidines in locally advanced rectal cancer (LARC). The study examined a radiomic model that included features extracted from magnetic resonance imaging (MRI) and contrast enhancement CT [49].

Five pathological complete response (pCR) prediction models: clinical; radiomic before and after ComBat; and combined before and after ComBat were evaluated, where ComBat is a statistical harmonization method called to deal with the “batch effect”. The addition of radiomic models that corrected inter-site variability and unbalanced data brings more predictive precision to the clinical model regarding the response to neoadjuvant treatment including fluropyrimidines [50,51,52].

With a sensitivity of 60%, a specificity of 77% and a sensitivity of 80%, a specificity of 69% and a negative predictive value (NPV) of 75% in cases of manual segmentation and automatic segmentation of tumor volumes, respectively, Defeudis and colleagues consider that automatic segmentation is superior in the use of radiomics as a biomarker from the point of view of sensitivity and NPV. The authors believe that radiomics will “pave the way” towards a personalization of treatment, both in the case of using the automatic segmentation method and the manual segmentation method of the tumor in radiotherapy planning [53].

The use of the MRI delta texture as a predictor of pCR and of the therapeutic results of locally advanced rectal cancer patients treated by chemoradiotherapy with capecitabine (825 mg/m^2^, twice daily five days a week) plus concurrent radiotherapy for 5 weeks and subsequently referred to radical surgery was proposed on a retrospective analysis including 100 patients. The MRI images used to delineate the target volumes included T2 DWI sequences and ADC sequences, and LifeX software was used to extract radiomic features. The textural delta variations were evaluated as a percentage between the pre-therapeutic values and the values obtained after the neoadjuvant treatment. The results identified a significant pCR correlation only with gray-level co-occurrence matrices (Co-GLCM)-Entropy extracted from ADC. The authors mention the possibility of using the method as a biomarker for selecting patients who will benefit from radical surgery after neoadjuvant treatment. The evaluation of the method in multicenter trials is necessary for the possible adoption of delta-radiomics in clinical practice [54].

MicroRNA (abbreviated miRNA) are small single-stranded non-coding RNA molecules that are involved in the regulation of gene expression and can modulate pathways that are associated with chemoresistance. The unfavorable response to capecitabine was correlated with miRNAs, especially in colorectal cancer cases that benefited from FOLFOX or XELOX regimens. Here, miRNA-19a demonstrated sensitivity of 66.7% and specificity of 63.9% to discriminate between FOLFOX-resistant colorectal cancer patients and FOLFOX-responsive cases. However, miRNA-19a was associated with the prediction of response to other chemotherapeutic agents (paclitaxel, epirubicin) and target therapies (gefitinib) in other cancers such as breast cancer and lung cancer. The miR-17-5p, a member of miR-17-92 cluster, is upregulated in FOLFOX-resistant colorectal cancer, and the reduced expression of serum exosomal miR-4772-3p is associated with early recurrence in stage II and stage III colorectal cancer after treatment with the FOLFOX regimen. In the case of head and neck cancers, Dai et al. mention the role of miRNAs in drug resistance by regulating apoptosis pathways, epithelial-mesenchymal transition (EMT) and cancer stem cells (CSCs) [55,56,57].

## 7. Conclusions

Capecitabine will certainly be part of the therapeutic protocols of HNC, both in induction settings in a concurrent approach with radiotherapy, but even in the palliative treatment of recurrent or metastatic disease including HNC re-irradiation. The identification of patient groups that will benefit from capecitabine more than 5-FU chemotherapy, both in terms of tumor control and the reduction of toxicities will be the object of future studies. An approach including routine testing of DPD or even the TP/DPD ratio and includes miRNAs, imaging and radiomics parameters in multi-omics biomarker models will help us implement a “precision chemotherapy” in HNC. This will help us avoid discontinuing or delaying treatment in this type of cancer. The chemosensitivity and prognostic features of HPV-OPC cancers open new horizons for the use of capecitabine in heavily pretreated metastatic cases. Vorinostat and lapatinib are agents that can be associated with capecitabine in future clinical trials to increase the therapeutic ratio.

## Data Availability

Not applicable.

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
