# Peer review of "Capecitabine—A “Permanent Mission” in Head and Neck Cancers “War Council”?"

_jcm, 2022, doi:10.3390/jcm11195582_

Round 1

Reviewer 1 Report

I would like to thank the Editor for the invitation to review this paper. The authors present an interesting review on the role of Capecitabine in the treatment of head and neck cancer. With respect to Iqbal’s review (Iqbal H, Pan Q. Capecitabine for treating head and neck cancer. Expert Opin Investig Drugs. 2016;25(7):851-859. doi:10.1080/13543784.2016.1181747), this paper provides little news apart from some clinical trial referring to biomarkers or radiomics. In addition, I miss more order in presenting the results in the different head and neck locations. Perhaps a review focused on the role of biomarkers and radiomics would be more interesting and novel. On the other hand, English writing must be improved.

Author Response

Dear Reviewer,

Thank you for the effort to analyze and evaluate this article and for the recommendations made. I added a mention in which I specified the database and the interval used, the evaluation criterion of the works being a subjective one, being a narrative review. I also corrected some expressions and sentences, especially in the abstract. I corrected the duplication of the bibliographic reference and added other references and new data about some aspects including HPV related head and neck cancers and associations of Capecitabine with other agents such as Vorinostat or Lapatinib. We also developed the radiomics and miRNAs chapter, bringing new data to advocate for the inclusion of Capecitabine in the treatment of head and neck cancer.

Hoping that you will appreciate these additions and revisions, we await new suggestions

Kind regards,

Camil Mirestean

Reviewer 2 Report

The work deals with the current topic, which is the search for new treatment options in patients with cancer of the head and neck region. The work is eligible after minor proofreading. It requires reviewing the literature, work number 23 is listed again as 26.

Author Response

(The authors gave the same response as above.)

Reviewer 3 Report

The authors chose a relevant and current topic. However, as it is a narrative review, it has some limitations:

The selection strategy of the manuscripts included in the review was not described, as well as the criteria for inclusion and exclusion of manuscripts.

Which electronic databases were used to select articles?

What was the search period?

Did the authors use any quality scale or analyze the risk of bias in the studies?

Although the text is very well written, we believe it is necessary to restructure the manuscript.

Author Response

(The authors gave the same response as above.)

Round 2

Reviewer 1 Report

I appreciate the changes made by the authors and do not consider further corrections necessary.